# PROGRESSIVE REINFORCEMENT LEARNING WITH DISTILLATION FOR MULTI-SKILLED MOTION CONTROL

**Glen Berseth\*, Cheng Xie\*, Paul Cernek, Michiel Van de Panne**
gberseth@cs.ubc.ca,cheng.k.xie@gmail.com,pcernek@cs.ubc.ca,
van@cs.ubc.ca
University of British Colubia

## ABSTRACT

Deep reinforcement learning has demonstrated increasing capabilities for continuous control problems, including agents that can move with skill and agility through their environment. An open problem in this setting is that of developing good strategies for integrating or merging policies for multiple skills, where each individual skill is a specialist in a specific skill and its associated state distribution. We extend policy distillation methods to the continuous action setting and leverage this technique to combine *expert* policies, as evaluated in the domain of simulated bipedal locomotion across different classes of terrain. We also introduce an *input injection* method for augmenting an existing policy network to exploit new input features. Lastly, our method uses transfer learning to assist in the efficient acquisition of new skills. The combination of these methods allows a policy to be incrementally augmented with new skills. We compare our progressive learning and integration via distillation (PLAID) method against three alternative baselines.

## 1 INTRODUCTION

As they gain experience, humans develop rich repertoires of motion skills that are useful in different contexts and environments. Recent advances in reinforcement learning provide an opportunity to understand how motion repertoires can best be learned, recalled, and augmented. Inspired by studies on the development and recall of movement patterns useful for different locomotion contexts (Roemmich & Bastian, 2015), we develop and evaluate an approach for learning multi-skilled movement repertoires. In what follows, we refer to the proposed method as PLAID: Progressive Learning and Integration via Distillation.

For *long lived* applications of complex control tasks a learning system may need to acquire and integrate additional skills. Accordingly, our problem is defined by the sequential acquisition and integration of new skills. Given an existing controller that is capable of one-or-more skills, we wish to: (a) efficiently learn a new skill or movement pattern in a way that is informed by the existing control policy, and (b) to reintegrate that into a single controller that is capable of the full motion repertoire. This process can then be repeated as necessary. We view PLAID as a continual learning method, in that we consider a context where all tasks are not known in advance and we wish to learn any new task in an efficient manner. However, it is also proves surprisingly effective as a multitask solution, given the three specific benchmarks that we compare against. In the process of acquiring a new skill, we also allow for a control policy to be augmented with additional inputs, without adversely impacting its performance. This is a process we refer to as *input injection*.

Understanding the time course of sensorimotor learning in human motor control is an open research problem (Wolpert & Flanagan, 2016) that exists concurrently with recent advances in deep reinforcement learning. Issues of generalization, context-dependent recall, transfer or "savings" in fast

---

\* These authors contributed equally to this work.

learning, forgetting, and scalability are all in play for both human motor control models and the learning curricula proposed in reinforcement learning. While the development of hierarchical models for skills offers one particular solution that supports scalability and that avoids problems related to forgetting, we eschew this approach in this work and instead investigate a progressive approach to integration into a control policy defined by a single deep network.

*Distillation* refers to the problem of combining the policies of one or more *experts* in order to create one single controller that can perform the tasks of a set of *experts*. It can be cast as a supervised regression problem where the objective is to learn a model that matches the output distributions of all *expert* policies (Parisotto et al., 2015; Teh et al., 2017; Rusu et al., 2015). However, given a new task for which an *expert* is not given, it is less clear how to learn the new task while successfully integrating this new skill in the pre-existing repertoire of the control policy for an agent. One well-known technique in machine learning to significantly improve sample efficiency across similar tasks is to use Transfer Learning (TL) (Pan & Yang, 2010), which seeks to reuse knowledge learned from solving a previous task to efficiently learn a new task. However, transferring knowledge from previous tasks to new tasks may not be straightforward; there can be *negative transfer* wherein a previously-trained model can take longer to learn a new task via fine-tuning than would a randomly-initialized model (Rajendran et al., 2015). Additionally, while learning a new skill, the control policy should not *forget* how to perform old skills.

The core contribution of this paper is a method Progressive Learning and Integration via Distillation (PLAiD) to repeatedly expand and integrate a motion control repertoire. The main building blocks consist of policy transfer and multi-task policy *distillation*, and the method is evaluated in the context of a continuous motor control problem, that of robust locomotion over distinct classes of terrain. We evaluate the method against three alternative baselines. We also introduce *input injection*, a convenient mechanism for adding inputs to control policies in support of new skills, while preserving existing capabilities.

## 2 RELATED WORK

Transfer learning and distillation are of broad interest in machine learning and RL (Pan & Yang, 2010; Taylor & Stone, 2009; Teh et al., 2017). Here we outline some of the most relevant work in the area of Deep Reinforcement Learning (DRL) for continuous control environments.

**Distillation** Recent works have explored the problem of combining multiple *expert* policies in the reinforcement learning setting. A popular approach uses supervised learning to combine each policy by regression over the action distribution. This approach yields model compression (Rusu et al., 2015) as well as a viable method for multi-task policy transfer (Parisotto et al., 2015) on discrete action domains including the Arcade Learning Environment (Bellemare et al., 2013). We adopt these techniques and extend them for the case of complex continuous action space tasks and make use of them as building block.

**Transfer Learning** Transfer learning exploits the structure learned from a previous task in learning a new task. Our focus here is on transfer learning in environments consisting of continuous control tasks. The concept of appending additional network structure while keeping the previous structure to reduce *catastrophic forgetting* has worked well on Atari games (Rusu et al., 2015; Parisotto et al., 2015; Rusu et al., 2016; Chen et al., 2015) Other methods reproduce data from all tasks to reduce the possibility of forgetting how to perform previously learned skills e.g, (Shin et al., 2017; Li & Hoiem, 2016). Recent work seeks to mitigate this issue using selective learning rates for specific network parameters (Kirkpatrick et al., 2017). A different approach to combining policies is to use a hierarchical structure (Tessler et al., 2016). In this setting, previously-learned policies are available as options to execute for a policy trained on a new task. However, this approach assumes that the new tasks will be at least a partial composition of previous tasks, and there is no reintegration of newly learned tasks. A recent promising approach has been to apply meta-learning to achieve control policies that can quickly adapt their behaviour according to current rewards (Finn et al., 2017). This work is demonstrated on parameterized task domains. The Powerplay method provides a general framework for training an increasingly general problem solver (Schmidhuber, 2011; Srivastava et al., 2012). It is based on iteratively: inventing a new task using play or invention; solving this task; and, lastly, demonstrating the ability to solve all the previous tasks. The last

two stages are broadly similar to our PLAID approach, although to the best of our knowledge, there are no experiments on motor control tasks of comparable complexity to the ones we tackle. In our work, we develop a specific progressive learning-and-distillation methodology for motor skills, and provide a detailed evaluation as compared to three other plausible baselines. We are specifically interested in understanding issues that arise from the interplay between transfer from related tasks and the forgetting that may occur.

**Hierarchical RL** further uses modularity to achieve transfer learning for robotic tasks (Tessler et al., 2016) This allows for the substitution of network modules for different robot types over a similar tasks (Devin et al., 2017). Other methods use Hierarchical Reinforcement Learning (HRL) as a method for simplifying a complex motor control problem, defining a decomposition of the overall task into smaller tasks (Kulkarni et al., 2016; Heess et al., 2016; Peng et al., 2017) While these methods examine knowledge transfer, they do not examine the reintegration of policies for related tasks and the associated problems such as *catastrophic forgetting*. Recent work examines learned motions that can be shaped by prior mocap clips (Merel et al., 2017), and that these can then be integrated in a hierarchical controller.

## 3 FRAMEWORK

In this section we outline the details of the Reinforcement Learning (RL) framework. We also give an introduction to the concepts of TL and *distillation*.

### 3.1 REINFORCEMENT LEARNING

Leveraging the framework of reinforcement learning, we frame the problem as a Markov Decision Processes (MDP): at each time step $t$, the world (including the agent) is in a state $s_t \in S$, wherein the agent is able to perform actions $a_t \in A$, sampled from a policy $\pi(s_t, a_t) = p(a_t|s_t)$ and resulting in state $s_{t+1} \in S$ according to transition probabilities $T(s_t, a_t, s_{t+1})$. Performing action $a_t$ from state $s_t$ produces a reward $r_t$; the expected cumulative reward earned from following some policy $\pi$ may then be written as:

$$J(\pi) = \mathbb{E}_{r_0,...,r_T} \left[ \sum_{t=0}^{T} \gamma^t r_t \right] \tag{1}$$

where $T$ is the time horizon, and $\gamma$ is the discount factor, defining the planning horizon length.

The agent's goal is to learn an optimal policy, $\pi^*$, maximizing $J(\pi)$. If the policy has parameters $\theta_\pi$, then the goal may be reformulated to identify the optimal parameters $\theta_\pi^*$:

$$\theta_\pi^* = \arg\max_{\theta_\pi} J(\pi(\cdot|\theta_\pi)) \tag{2}$$

Our policy models a Gaussian distribution with a mean state dependent mean, $\mu_{\theta_t}(s_t)$. Thus, our stochastic policy may be formulated as follows:

$$a_t \sim \pi(a_t \mid s_t, \theta_\pi) = \mathcal{N}(\mu(s_t \mid \theta_\mu), \Sigma) \qquad \Sigma = diag\{\sigma_i^2\} \tag{3}$$

where $\Sigma$ is a diagonal covariance matrix with entries $\sigma_i^2$ on the diagonal, similar to (Peng et al., 2017).

To optimize our policy, we use stochastic policy gradient methods, which are well-established family of techniques for reinforcement learning (Sutton et al., 2000). The gradient of the expected reward with respect to the policy parameters, $\nabla_{\theta_\pi} J(\pi(\cdot|\theta_\pi))$, is given by:

$$\nabla_{\theta_\pi} J(\pi(\cdot|\theta_\pi)) = \int_S d_\theta(s) \int_A \nabla_{\theta_\pi} \log(\pi(a, s|\theta_\pi)) A_\pi(s, a) \, da \, ds \tag{4}$$

where $d_\theta = \int_S \sum_{t=0}^{T} \gamma^t p_0(s_0)(s_0 \to s \mid t, \pi_0) \, ds_0$ is the discounted state distribution, $p_0(s)$ represents the initial state distribution, and $p_0(s_0)(s_0 \to s \mid t, \pi_0)$ models the likelihood of reaching

state $s$ by starting at state $s_0$ and following the policy $\pi(a, s|\theta_\pi)$ for $T$ steps (Silver et al., 2014). $A_\pi(s, a)$ represents an advantage function (Schulman et al., 2016). In this work, we use the Positive Temporal Difference (PTD) update proposed by (Van Hasselt, 2012) for $A_\pi(s, a)$:

$$A_\pi(s_t, a_t) = I\left[\delta_t > 0\right] = \begin{cases} 1, & \delta_t > 0 \\ 0, & \text{otherwise} \end{cases} \tag{5}$$

$$\delta_t = r_t + \gamma V_\pi(s_{t+1}) - V_\pi(s_t) \tag{6}$$

where $V_\pi(s) = \mathbb{E}\left[\sum_{t=0}^{T} \gamma^t r_t \mid s_0 = s\right]$ is the value function, which gives the expected discounted cumulative reward from following policy $\pi$ starting in state $s$. PTD has the benefit of being insensitive to the advantage function scale. Furthermore, limiting policy updates in this way to be only in the direction of actions that have a positive advantage has been found to increase the stability of learning (Van Hasselt, 2012). Because the true value function is unknown, an approximation $V_\pi(\cdot \mid \theta_v)$ with parameters $\theta_v$ is learned, which is formulated as the regression problem:

$$\underset{s_t, r_t, s_{t+1}}{\text{minimize}} \, \mathbb{E}\left[\frac{1}{2}\left(y_t - V_\pi(s \mid \theta_v)\right)^2\right], \qquad y_t = r_t + \gamma V_\pi(s_{t+1} \mid \theta_v) \tag{7}$$

## 3.2 POLICY DISTILLATION

Given a set of *expert* agents that have solved/mastered different tasks we may want to combine the skills of these different *experts* into a single multi-skilled agent. This process is referred to as *distillation*. *Distillation* does not necessarily produce an optimal mix of the given *experts* but instead tries to produce an *expert* that best matches the action distributions produced by all *experts*. This method functions independent of the reward functions used to train each *expert*. *Distillation* also scales well with respect to the number of tasks or *experts* that are being combined.

## 3.3 TRANSFER LEARNING

Given an *expert* that has solved/mastered a task we want to reuse that *expert* knowledge in order to learn a new task efficiently. This problem falls in the area of *Transfer Learning* (Pan & Yang, 2010). Considering the state distribution *expert* is skilled at solving, ($D_{\omega_i}$ the *source* distribution) it can be advantageous to start learning a *new*, target task $\omega_{i+1}$ with *target* distribution $D_{\omega_{i+1}}$ using assistance from the *expert*. The agent learning how to solve the *target* task with domain $D_{\omega_{i+1}}$ is referred to as the *student*. When the *expert* is used to assist the *student* in learning the target task it can be referred to as the *teacher*. The success of these methods are dependent on overlap between the $D_{\omega_i}$ and $D_{\omega_{i+1}}$ state distributions.

## 4 PROGRESSIVE LEARNING

Although we focus on the problem of being presented with tasks sequentially, there exist other methods for learning a multi-skilled character. We considered 4 overall integration methods for learning multiple skills, the first being a controller that learns multiple tasks at the same time (MultiTasker), where a number of skills are learned at the same time. It has been shown that learning many tasks together can be faster than learning each task separately (Parisotto et al., 2015). The curriculum for using this method is shown in Figure 1a were during a single RL simulation all tasks are learned together. It is also possible to randomly initialize controllers and train in parallel (Parallel) and then combine the resulting policies Figure 1b. We found that learning many skills from scratch was challenging, we were only able to get fair results for the *flat* task. Also, when a *new* task is to be learned with the Parallel model it would occur outside of the original parallel learning, leading to a more sequential method. A TL-Only method that uses TL while learning tasks in a sequence Figure 1c, possibly ending with a *distillation* step to combine the learned policies to decrease forgetting. For more details see Appendix: 8.4. The last version (PLAiD) learns each task sequentially using TL from the previous, most skilled policy, in the end resulting in a policy capable of solving all tasks Figure 1d. This method works well for both combining learned skills and learning new skills.

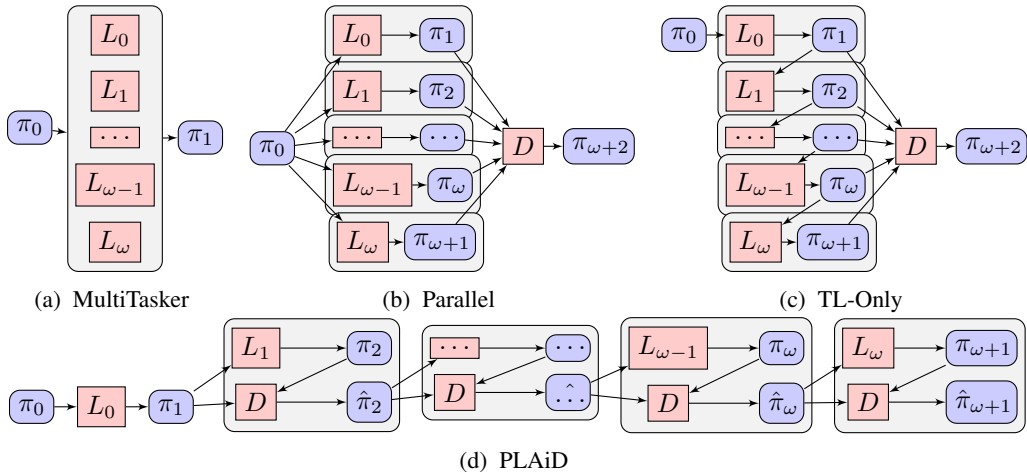

Figure 1: Different curriculum learning process. The red box with a $D$ in it denotes a *distillation* step that combines policies. Each gray box denotes one iteration of learning a new policy. The larger red boxes with an $L_{terrain-type}$ denotes a learning step where a new skill is learned.

## 4.1 PROGRESSIVE LEARNING AND INTEGRATION VIA DISTILLATION

In this section, we detail our proposed learning framework for continual policy transfer and *distillation* (PLAiD). In the acquisition (TL) step, we are interested in learning a new task $\omega_{i+1}$. Here transfer can be beneficial if the task structure is somewhat similar to previous tasks $\omega_i$. We adopt the TL strategy of using an existing policy network and fine-tuning it to a new task. Since we are not concerned with retaining previous skills in this step, we can update this policy without concern for forgetting. As the agent learns it will develop more skills and the addition of every new skill can increase the probability of transferring knowledge to assist the learning of the next skill.

In the integration (*distillation*) step, we are interested in combining all past skills $(\pi_0, \ldots, \pi_i)$ with the newly acquired skill $\pi_{i+1}$. Traditional approaches have used policy regression where data is generated by collecting trajectories of the *expert* policy on a task. Training the *student* on these trajectories does not always result in robust behaviour. This poor behaviour is caused by the *student* experiencing a different distribution of trajectories than the *expert* during evaluation. To compensate for this distribution difference, portions of the trajectories should be generated by the *student*. This allows the *expert* to suggest behaviour that will pull the state distribution of the *student* closer to the *expert*'s. This is a common problem in learning a model to reproduce a given distribution of trajectories (Ross et al., 2010; Bengio et al., 2015; Martinez et al., 2017; Lamb et al., 2016). We use a method similar to the DAGGER algorithm (Ross et al., 2010) which is useful for distilling policies (Parisotto et al., 2015). See Appendix: 8.2.1 for more details. As our RL algorithm is an actor-critic method, we also perform regression on the critic by fitting both in the same step.

## 4.2 HIGH LEVEL EXPERIMENT DESIGN

The results presented in this work cover a range of tasks that share a similar action space and state space. Our focus is to demonstrate continual learning between related tasks. In addition, the conceptual framework allows for extensions that would permit differing state spaces, described later in Section: 5.2.

## 5 RESULTS

In this experiment, our set of tasks consists of 5 different terrains that a 2D humanoid walker (*pd-biped*) learns to traverse. The humanoid walker is trained to navigate multiple types of terrain including *flat* in (Figure 6a), *incline* (Figure 6b), *steps* (Figure 6c), *slopes* (Figure 6d), *gaps* (Figure 6e) and a combination of all terrains *mixed* (Figure 6f) on which agents are trained. The goal in these tasks is to maintain a consistent forward velocity traversing various terrains, while also

matching a motion capture clip of a natural human walking gait on flat ground, similar to (Peng & van de Panne, 2016). The *pd-biped* receives as input both a character and (eventually) a terrain state representation, consisting of the terrains heights of 50 equally-spaced points in front of the character. The action space is 11-dimensional, corresponding to the joints. Reasonable torque limits are applied, which helps produce more natural motions and makes the control problem more difficult. A detailed description of the experimental setup is included in Section: 8.5. The tasks are presented to the agent sequentially and the goal is to progressively learn to traverse all terrain types.

We evaluate our approach against three baselines. First, we compare the above learning curriculum from learning new tasks in PLAiD with learning new tasks in Parallel. This will demonstrate that knowledge from previous tasks can be effectively transferred after *distillation* steps. Second, we compare to the MultiTasker to demonstrate that iterated *distillation* is effective for the retention of learned skills. The MultiTasker is also used as a baseline for comparing learning speed. Last, a method that performs TL between tasks and concludes with a *distillation* step is evaluated to illustrate the result of different TL and *distillation* schedules. The results of the PLAiD controller are displayed in the accompanying Video [1]

## 5.1 TRANSFER LEARNING

First, the *pd-biped* is trained to produce a walking motion on flat ground (*flat*). In Figure 2a PLAiD is compared to the three baselines for training on *incline*. The TL-Only method learns fast as it is given significant information about how to perform similar skills. The Parallel method is given no prior information leading to a less skilled policy. The first MultiTasker for the *incline* task is initialized from a terrain injected controller that was trained to walk on *flat* ground. Any subsequent MultiTasker is initialized from the final MultiTasker model of the preceding task. This controller has to learn multiple tasks together, which can complicate the learning process, as simulation for each task is split across the training and the overall RL task can be challenging. This is in contrast to using PLAiD, that is also initialized with the same policy trained on *flat*, that will integrate skills together after each new skill is learned.

In Figure 2b the MultiTasker is learning the new task (*steps*) with similar speed to PLAiD. However, after adding more tasks the MultiTasker is beginning to struggle in Figure 2c and starts to *forget* in Figure 2d, with the number of tasks it must learn at the same time. While PLAiD learns the new tasks faster and is able to integrate the new skill required to solve the task robustly. TL-Only is also able to learn the new tasks very efficiently.

## 5.2 INPUT FEATURE INJECTION

An appealing property of using *distillation* in PLAiD is that the combined policy model need not resemble that of the individual *expert* controllers. For example, two different *experts* lacking state features and trained without a local map of the terrain can be combined into a single policy that has new state features for the terrain. These new terrain features can assist the agent in the task domain in which it operates.

We introduce the idea of *input injection* for this purpose. We augment a policy with additional input features while allowing it to retain its original functional behaviour similar to (Chen et al., 2015). This is achieved by adding additional inputs to the neural network and initializing the connecting layer weights and biases to 0. By only setting the weights and biases in the layer connecting the new features to the original network to 0, the gradient can still propagate to any lower layers which are initialized random without changing the functional behaviour. This is performed when distilling the *flat* and *incline experts*. Further details can be found in Appendix: 8.3.

## 5.3 DISTILLING MULTIPLE POLICIES

Training over multiple tasks at the same time may help the agent learn skills quicker, but this may not scale with respect to the number of tasks. When training the MultiTasker over two or even three tasks (Figure 3a) the method displays good results, however when learning a fourth or more tasks the method struggles, as shown in Figure 3b and 3b. Part of the reason for this struggle is when new

---

[1] $https://youtu.be/_DjHbHCXGk0$

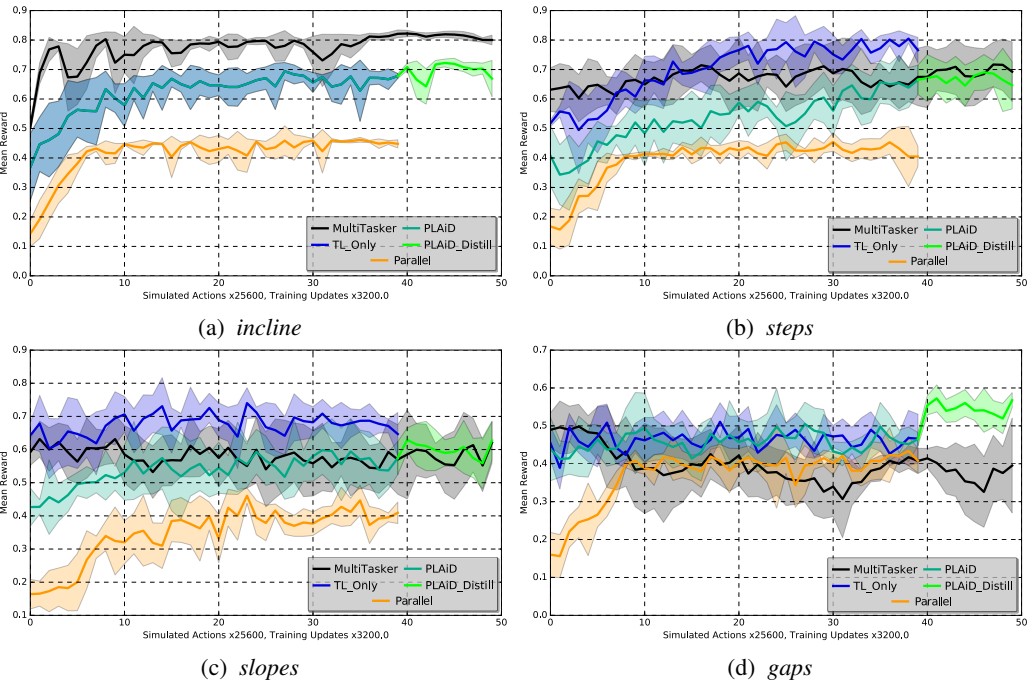

Figure 2: Learning comparison over each of the environments. These plots show the mean and std over 5 simulations, each initialized with different random seeds. The learning for PLAiD is split into two steps, with TL (in green) going first followed by the *distillation* part (in yellow).

| Tasks | flat | incline | steps | slopes | gaps | average |
|---|---|---|---|---|---|---|
| PLAiD | 0.054 | 0.155 | 0.001 | 0.043 | −0.083 | 0.063 |
| TL-Only | −0.065 | −0.044 | −0.235 | −0.242 | 0.000 | −0.147 |
| TL-Only (with Distill) | 0.068 | 0.039 | −0.030 | −0.062 | −0.133 | −0.024 |
| MultiTasker | −0.001 | −0.053 | −0.030 | 0.119 | 0.000 | 0.009 |

Table 1: These values are relative percentage changes in the average reward, where a value of $0$ is no forgetting and a value of $−1$ corresponds to completely forgetting how to perform the task. A value $> 0$ corresponds to the agent learning how to better perform a task after training on other tasks. Here, the final policy after training on *gaps* compared to the original polices produced at the end of training for the task noted in the column heading. The TL-Only baseline forgets more than PLAiD. The MultiTasker forgets less than PLAiD but has a lower average reward over the tasks.

tasks are added the MultiTasker has to make trade-offs between more tasks to maximizes. As more tasks are added, this trade-off becomes increasingly complex resulting in the MultiTasker favouring easier tasks. Using PLAiD to combine the skills of many policies appears to scale better with respect to the number of skills being integrated. This is likely because distillation is a semi-supervised method which is more stable than the un-supervised RL solution. This can be seen in Figure 3d, 3e and especially in 3f where PLAiD combines the skills faster and can find higher value policies in practice. PLAiD also presents zero-shot training on tasks which it has never been trained on. In Figure 7 this generalization is shown as the agent navigates across the *mixed* environment.

This is also reflected in Table 1, that shows the final average reward when comparing methods before and after distillation. The TL-Only is able to achieve high performance but much is lost when learning new tasks. A final distillation step helps mitigate this issue but does not work as well as PLAiD. It is possible performing a large final *distillation* step can lead to over-fitting.

There are some indications that *distillation* is hindering training during the initial few iterations. We are initializing the network used in *distillation* with the most recently learning policy after TL. The large change in the initial state distribution from the previous seen distribution during TL could

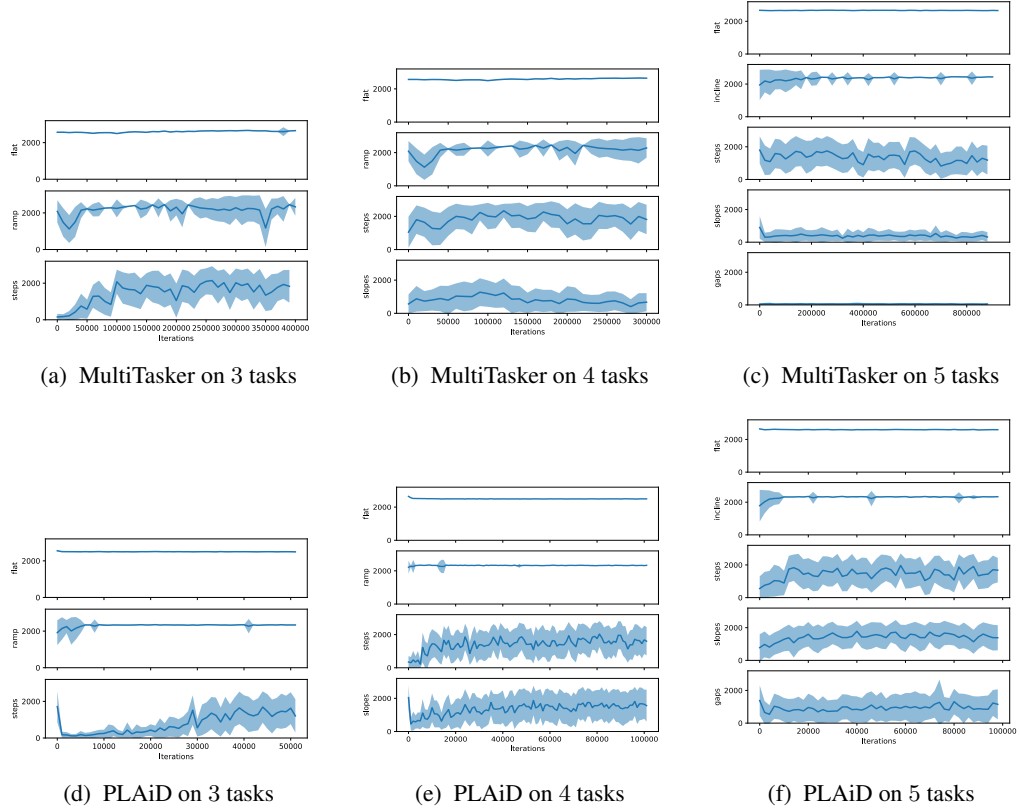

(a) MultiTasker on 3 tasks      (b) MultiTasker on 4 tasks      (c) MultiTasker on 5 tasks

(d) PLAiD on 3 tasks      (e) PLAiD on 4 tasks      (f) PLAiD on 5 tasks

Figure 3: These figures show the average reward a particular policy achieves over a number of tasks.

be causing larger gradients to appear, disrupting some of the structure learned during the TL step, shown in Figure 3d and 3e. There also might not exist a smooth transition in policy space between the newly learned policy and the previous policy distribution.

## 6 DISCUSSION

**MultiTasker vs PLAiD:**     The MultiTasker may be able to produce a policy that has higher overall average reward, but in practise constraints can keep the method from combining skills gracefully. If the reward functions are different between tasks, the MultiTasker can favour a task with higher rewards, as these tasks may receive higher advantage. It is also a non-trivial task to normalize the reward functions for each task in order to combine them. The MultiTasker may also favour tasks that are easier than other tasks in general. We have shown that the PLAiD scales better with respect to the number of tasks than the MultiTasker. We expect PLAiD would further outperform the MultiTasker if the tasks were more difficult and the reward functions dissimilar.

In our evaluation we compare the number of iterations PLAiD uses to the number the MultiTasker uses on only the *new* task, which is not necessarily fair. The MultiTasker gains its benefits from training on the other tasks together. If the idea is to reduce the number of simulation samples that are needed to learn *new* tasks then the MultiTasker would fall far behind. *Distillation* is also very efficient with respect to the number of simulation steps needed. Data could be collected from the simulator in groups and learned from in many batches before more data is needed as is common for behavioural cloning. We expect another reason *distillation* benefits learning multiple tasks is that the integration process assists in pulling policies out of the local minima RL is prone to.

**Transfer Learning:**     Because we are using an actor-critic learning method, we also studied the possibility of using the value functions for TL. We did not discover any empirical evidence that this

assisted the learning process. When transferring to a new task, the state distribution has changed and the reward function may be completely different. This makes it unlikely that the value function will be accurate on this new task. In addition, value functions are in general easier and faster to learn than policies, implying that value function reuse is less important to transfer. We also find that helpfulness of TL depends on not only the task difficulty but the reward function as well. Two tasks may overlap in state space but the area they overlap could be easily reachable. In this case TL may not give significant benefit because the overall RL problem is easy. The greatest benefit is gained from TL when the state space that overlaps for two tasks is difficult to reach and in that difficult to reach area is where the highest rewards are achieved.

## 6.1 LIMITATIONS:

Once integrated, the skills for our locomotion tasks are self-selecting based on their context, i.e., the knowledge of the upcoming terrain. It may be that other augmentation and distillation strategies are better for situations where either the reward functions are different or a one-hot vector is used to select the currently active expert. In our transfer learning results we could be over fitting the initial *expert* for the particular task it was learning. Making it more challenging for the policy to learn a new task, resulting in *negative transfer*. After learning many new tasks the previous tasks may not receive a large enough potion of the *distillation* training process to preserve the *experts* skill well enough. How best to chose which data should be trained on next to best preserve the behaviour of *experts* is a general problem with multi-task learning. *Distillation* treats all tasks equally independent of their reward. This can result in very low value tasks, receiving potentially more distribution than desired and high value tasks receiving not enough. We have not needed the use a *one-hot* vector to indicate what task the agent is performing. We want the agent to be able to recognize which task it is given but we do realize that some tasks could be too similar to differentiate, such as, walking vs jogging on flat ground.

## 6.2 FUTURE WORK:

It would be interesting to develop a method to prioritize tasks during the distillation step. This could assist the agent with forgetting issues or help with relearning tasks. While we currently use the Mean Squared Error (MSE) to pull the distributions of *student* policies in line with *expert* polices for *distillation*, better distance metrics would likely be helpful. Previous methods have used KL Divergence in the discrete action space domain where the state-action value function encodes the policy, e.g., as with Deep Q-Network (DQN). In this work we do not focus on producing the best policy from a *mixture of experts*, but instead we match the distributions from a number of *experts*. The difference is subtle but in practice it can be more challengine to balance many *experts* with respect to their reward functions. It could also be beneficial to use a KL penalty while performing *distillation*, i.e., something similar to the work in (Teh et al., 2017) in order to keep the policy from changing too rapidly during training.

## 7 CONCLUSION

We have proposed and evaluated a method for the progressive learning and integration (via distillation) of motion skills. The method exploits transfer learning to speed learning of new skills, along with input injection where needed, as well as continuous-action *distillation*, using DAGGER-style learning. This compares favorably to baselines consisting of learning all skills together, or learning all the skills individually before integration. We believe that there remains much to learned about the best training and integration methods for movement skill repertoires, as is also reflected in the human motor learning literature.

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

# 8 APPENDIX

## 8.1 NETWORK MODELS

We used two different Network models for the experiments in this paper. The first model is a *blind* model that does not have any terrain features. The *blind* policy is a Neural Network with 2 hidden layers ($512 \times 256$) with ReLU activations. The output layer of the policy network has linear activations. The network used for the value function has the same design except there is 1 output on the final layer. This design is used for the *flat* and *incline* tasks.

We augment the *blind* network design by adding features for terrain to create an agent with *sight*. This network with *terrain features* has a single convolution layer with 8 filters of width 3. This constitutional layer is followed by a dense layer of 32 units. The dense layer is then concatenated twice, once along each of the original two hidden layers in the *blind* version of the policy.

## 8.2 Hyper Parameters and Training

The policy network models a Gaussian distribution by outputting a state dependant mean. We use a state independent standard deviation that normalized with respect to the action space and multiplied by 0.1. We also use a version of epsilon greedy exploration where with $\epsilon$ probability an exploration action is generated. For all of our experiments we linearly anneal $\epsilon$ from 0.2 to 0.1 in 100,000 iterations and leave it from that point on. Each training simulation takes approximately 5 hours across 8 threads. For network training we use Stochastic Gradient Decent (SGD) with momentum. During the distillation step we use gradually anneal the probability of selecting an *expert* action from 1 to 0 over 10,000 iterations.

For the evaluation of each model on a particular task we use the average reward achieved by the agent over at most 100 seconds of simulation time. We average this over running the agent over a number of randomly generated simulation runs.

### 8.2.1 Distillation

For each of the distillation steps we initialize the policy from the most recently trained policy. This policy has seen all of the tasks thus far but may have overfit the most recent tasks. We us a version of the DAGGER algorithm for the distillation process (Ross et al., 2010). We anneal from selecting actions from the *expert* polices to selecting actions from the *student* policy The probability of selecting an action from the *expert* is annealed to near zero after 10,000 training updates. We still add exploration noise to the policies when generating actions to take in the simulation. This is also annealed along with the probability of selecting from the *expert* policy. The actions used for training always come from the *expert* policy. Although some actions are applied in the simulation from the *student*, during a training update those actions will be replaced with ones from the proper *expert*. The *expert* used to generate actions for tasks $0 - i$ is $\pi_i$ and the *expert* used to generate action for task $i + 1$ is $\pi_{i+1}$. We keep around at most 2 policies at any time.

## 8.3 Input Features and Injection

In order to add additional input features to the policy network we construct a new network. This new network has a portion of it that is the same design as the previous network plus additional parameters. First we initialize the new network with random parameters. Then we copy over the values from the previous network into the new one for the portion of the network design that matches the old. Then the weight for the layers that connect the old portion of the network to the new are set to 0. This will allow the network to preserve the previous distribution it modeled. Having the parameters from the old network will also help generate gradients to train the new 0 valued network parameters. We use feature injection to assist the learning method with differentiating between different states. For example, it could be challenging to discover the difference between the *flat* and *incline* tasks using only the character features. Therefore, we add new terrain features to allow the controller to better differentiate between these two different tasks.

## 8.4 TL-Only Baseline

We also evaluate a baseline where we TL for all tasks. In this baseline TL is performed for a number of tasks and then *distillation* is used to combined these many learned skills. This method can be considered a version of PLAiD where tasks are learned in groups and after some number of tasks, a collection of policies/skills are distilled together. In Figure 5 the learning curves for the TL-Only baseline are given. The TL-Only method learns new tasks well. We do not show the *incline* tasks as the two methods are the same up to starting the *steps* tasks. In Table 1 the amount of forgetting is compared between methods. To compare the amount of forgetting between TL-Only and PLAiD we show the relative loss in average reward between the original policy trained for the tasks *steps* and *slopes* and the final polices for each method on *gaps*. The TL-Only method shows a larger drop in

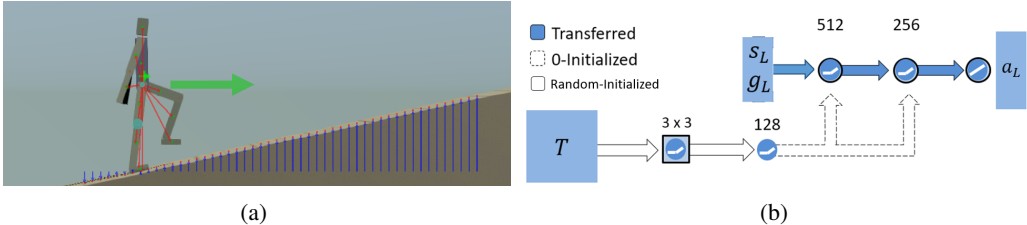

(a)               (b)

Figure 4: (a) The input features include both the character state shown as the red lines between the root of the character and the character's links and the terrain features shown as the blue arrows along the ground. (b) A diagram of method used to inject additional state features for the terrain.

| Tasks | flat | incline | steps | slopes | gaps | **average** |
|---|---|---|---|---|---|---|
| PLAiD | 0.891 | 0.800 | 0.666 | 0.602 | 0.529 | 0.698 |
| TL-Only | 0.790 | 0.662 | 0.615 | 0.543 | 0.626 | 0.647 |
| TL-Only (with Distill) | 0.903 | 0.719 | 0.781 | 0.671 | 0.543 | 0.723 |
| MultiTasker | 0.844 | 0.757 | 0.677 | 0.656 | 0.504 | 0.688 |

Table 2: Final average reward for each method. Higher is better. Here, the final policy is after training on *gaps*. the PLAiD method achieves on average higher values across tasks.

policy performance corresponding to a large amount of forgeting compared to PLAiD, in particular for the more complex tasks *steps* and *slopes*. Interestingly, the final distllation step for TL-Only appears to reduce the performance of the policy. We believe this is related to the final distillation step being more challenging than performing a simpler distillation after each new task. Note that we only compare these two steps because the process for the first two tasks for PLAiD and TL-Only are the same. A comparison of the average rewards for the final policies are given in Table 2.

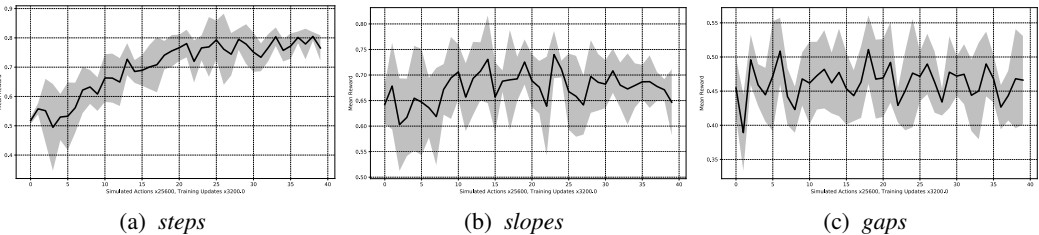

(a) *steps*          (b) *slopes*          (c) *gaps*

Figure 5: Transfer learning only baselines for each of the new tasks.

## 8.5   AGENT DESIGN

The agent used in the simulation models the dimensions and masses of the average adult. The size of the character state is 50 parameters that include the relative position and velocity of the links in the agent (Figure 4a). The action space consists of 11 parameters that indicate target joint positions for the agent. The target joint positions (pd-targets) are turned into joint torques via proportional derivative controllers at each joint.

The reward function for the agent consists of 3 primary terms. The first is a velocity term the rewards the agent for going at velocity of 1 m/s The second term is the difference between the pose of the agent and the current pose of a kinematic character controlled via a motion capture clip. The difference between the agent and the clip consists of the rotational difference between each corresponding joint and the difference in angular velocity. The angular velocity for the clip is approximated via finite differences between the current pose of the clip and it's last pose. The last term is an L2 penalty on the torques generated by the agent to help reduce spastic motions. We also impose torque limits on the joints to reduce unrealistic behaviour, limits: Hips 150, knees 125, ankles 100, shoulders 100, elbows 75 and neck 50 N/m.

**Terrain Types**  All terrain types are randomly generated per episode, except for the *flat* terrain. The *incline* terrain is slanted and the slant of the terrain is randomly sampled between 20 and 25 degrees. The *steps* terrain consists of flat segments with widths randomly sampled from 1.0 m to 1.5 m followed by sharp steps that have randomly generated heights between 5 cm and 15 cm. The *slopes* terrain is randomly generated by updating the slope of the previous point in the ground with a value sampled from −20 and 20 degrees to generate a new portion of the ground every 10 cm. The *gaps* terrain generate gaps of width 25 - 30 cm separated by flat segments of widths sampled from 2.0 m to 2.5 m. The *mixed* terrain is a combination of the above terrains where a portion is randomly chosen from the above terrain types.

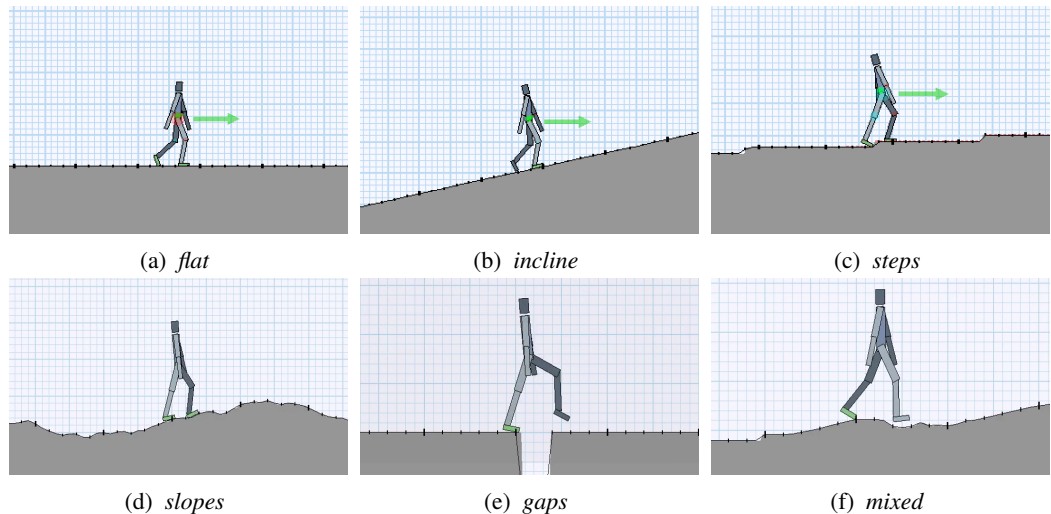

(a) *flat*          (b) *incline*          (c) *steps*

(d) *slopes*          (e) *gaps*          (f) *mixed*

Figure 6: The environments used to evaluate PLAiD.

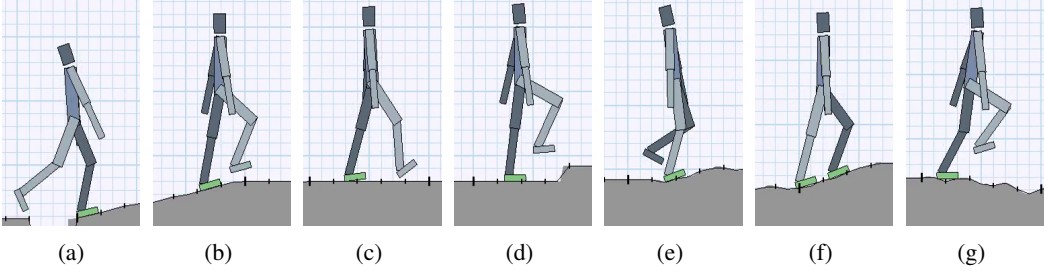

(a)          (b)          (c)          (d)          (e)          (f)          (g)

Figure 7: Still frame shots of the *pd-biped* traversing the *mixed* environment.

## 8.6 MULTITASKER

In certain cases the MultiTasker can learn new task faster than PLAiD. In Figure 8a we present the MultiTasker and compare it to PLAiD. In this case the MultiTasker splits its training time across multiple tasks, here we compare the two methods with respect to the time spent learning on the single *new* task. This is a good baseline to compare our method against but in some ways this is not fair. If the real measure of how efficient a learning method is the number of simulation samples that are needed to learn would fall far behind as the MultiTasker needs to train across all tasks to gain the benefits of improving a single task without forgetting the old tasks.

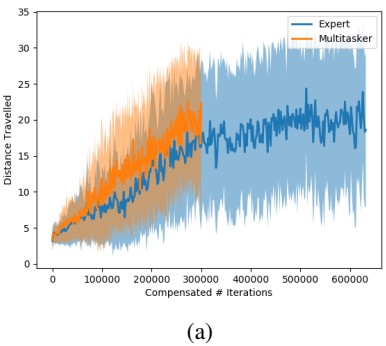

(a)

Figure 8: (a) Shows that the MultiTasker can learn faster on *steps*, *flat* and *incline* than PLAiD (expert) learning the single task *steps* with TL.

