# OpenReview forum: "Progressive Reinforcement Learning with Distillation for Multi-Skilled Motion Control"
_ICLR.cc/2018/Conference — Accept (Poster)_

### Official Review · AnonReviewer3 · 2017-11-25
**Good approach, needs more details.**

**Rating:** 7
**Confidence:** 4

**Review:**

This paper aims to learn a single policy that can perform a variety of tasks that were experienced sequentially. The approach is to learn a policy for task 1, then for each task k+1: copy distilled policy that can perform tasks 1-k, finetune to task k+1, and distill again with the additional task. The results show that this PLAID algorithm outperforms a network trained on all tasks simultaneously.

Questions:
- When distilling the policies, do you start from a randomly initialized policy, or do you start from the expert policy network?
- What data do you use for the distillation? Section 4.1 states"We use a method similar to the DAGGER algorithm", but what is your method. If you generate trajectories form the student network, and label them with the expert actions, does that mean all previous expert policies need to be kept in memory?
- I do not understand the purpose of "input injection" nor where it is used in the paper.

Strengths:
- The method is simple but novel. The results support the method's utility.
- The testbed is nice; the tasks seem significantly different from each other. It seems that no reward shaping is used.
- Figure 3 is helpful for understanding the advantage of PLAID vs MultiTasker.

Weaknesses:
- Figure 2: the plots are too small.
- Distilling may hurt performance ( Figure 2.d)
- The method lacks details (see Questions above)
- No comparisons with prior work are provided. The paper cites many previous approaches to this but does not compare against any of them.
- A second testbed (such as navigation or manipulation) would bring the paper up a notch.

In conclusion, the paper's approach to multitask learning is a clever combination of prior work. The method is clear but not precisely described. The results are promising. I think that this is a good approach to the problem that could be used in real-world scenarios. With some filling out, this could be a great paper.

---

> ### Public Comment · (anonymous) · 2017-12-28
> **Thank you for the interest, questions, and suggestions!**
>
> We believe that there is much to be explored for progressive learning and distillation of continuous action tasks, as exemplified by our control problems.
>
> Re:  during distillation, do we start from random policy or expert policy?
> The networks were initialized from the most recently trained policy, i.e., the one trained on the new task.
>
> Re: data used for distillation;  do all previous expert policies need to be kept in memory?
> We have added the following paragraph to the paper Appendix to address this.
>    For each of the distillation steps we initialize the policy from the most recently trained policy. This policy has seen all of the tasks thus far but may have overfit the most recent tasks. We us a version of the DAGGER algorithm for the distillation process (Ross et al., 2010). We anneal from select- ing actions from the expert policies to selecting actions from the student policy The probability of selecting an action from the expert is annealed to near zero after 10, 000 training updates. We still add exploration noise to the policies when generating actions to take in the simulation. This is also annealed along with the probability of selecting from the expert policy. The actions used for training always come from the expert policy. Although some actions are applied in the simulation from the student, during a training update those actions will be replaced with ones from the proper expert. The expert used to generate actions for tasks 0 − i is πi and the expert used to generate action for task i + 1 is πi+1. We keep around at most 2 policies at any time.
>
> Re: purpose of input injection
> We have added further details and explanations to the paper. Specifically, the “flat” walking is not provided with information about the upcoming terrain, while other policies (e.g., incline, steps, slopes, gaps) are provided with extra inputs (a linear height map) of the upcoming terrain.  Input injection allows for the “flat” walking policy to be used as the starting point for policies that have these additional inputs.
>
>
>
> Re: comparisons with prior work
> We compare the PLAID method to three other baselines, two of which are in the paper (MultiTasker, Parallel-Learn-then-Distill), and an additional baseline that we will soon have completed in response to the feedback (Successive Transfers then Distill).
> In what follows below, we comment further on other specific previous work.
> Progressive nets: While this is only tested on discrete actions, the idea itself is orthogonal to this issue. However, the existing baselines using DeepRL are not very applicable for different reasons. For the progressive net, you will get a set of experts in one net but the net itself does not know which expert to choose when it is given a task, i.e., which head of the network to choose.
> Attend Adapt and Transfer:  This method explains how to combine K experts in learning task T_i but it is not obvious how it would extend to learning T_i+1.
> Lifelong Learning in Minecraft: This relies on options with clear end definitions in sequential tasks. It is not directly obvious how this could be applied to our domain with continuously varying terrain and continuous actions.
> Distral:  This trains multiple policies in parallel, rather than one policy continually, and so it is largely captured by one of our baselines. Also, we note that the method is specific to discrete actions in its KL regularization term.

---

> ### Comment · AnonReviewer3 · 2018-01-13
> **Revision to review**
>
> I find the additions to the paper satisfactory and will increase my score accordingly.

---

### Official Review · AnonReviewer1 · 2017-11-27
**Nice continual learning study**

**Rating:** 7
**Confidence:** 4

**Review:**

This paper describes PLAID, a method for sequential learning and consolidation of behaviours via policy distillation; the proposed method is evaluated in the context of bipedal motor control across several terrain types, which follow a natural curriculum.

Pros:
- PLAID masters several distinct tasks in sequence, building up “skills” by learning “related” tasks of increasing difficulty.
- Although the main focus of this paper is on continual learning of “related” tasks, the authors acknowledge this limitation and convincingly argue for the chosen task domain.

Cons:
- PLAID seems designed to work with task curricula, or sequences of deeply related tasks; for this regime, classical transfer learning approaches are known to work well (e.g finetunning), and it is not clear whether the method is applicable beyond this well understood case.
- Are the experiments single runs? Due to the high amount of variance in single RL experiments it is recommended to perform several re-runs and argue about mean behaviour.

Clarifications:
- What is the zero-shot performance of policies learned on the first few tasks, when tested directly on subsequent tasks?
- How were the network architecture and network size chosen, especially for the multitasker? Would policies generalize to later tasks better with larger, or smaller networks?
- Was any kind of regularization used, how does it influence task performance vs. transfer?
- I find figure 1 (c) somewhat confusing. Is performance maintained only on the last 2 tasks, or all previously seen tasks? That’s what the figure suggests at first glance, but that’s a different goal compared to the learning strategies described in figures 1 (a) and (b).

---

> ### Public Comment · (anonymous) · 2017-12-28
> **Thanks for the review. We appreciate the various comments and questions!**
>
> Our work is (to the best of our knowledge) among the first to study, with a detailed evaluation, multi-task and continual learning on problems with continuous action spaces using deep learning.
>
>
> Re: for sequences of related tasks, transfer learning, e.g., fine-tuning, is “known to work well”
> ⇒ We are computing an additional “Successive Transfers then Distill” benchmark to show that this is not the case here. This learns the tasks, in sequence, followed by a final distillation step, and thus without the progressive intermediate distillation steps used in PLAID. Our current results for this benchmark show a significant benefit for PLAID.  For comments regarding other potential baselines, please also see our replies to the other reviewers.
>
> Re: single runs
> Yes, currently these are single runs. We are re-running all simulations 5 times in order to provide more sound comparisons regarding the relative merit of the methods. We will post these in the coming days. There are no surprises in the results to date.
>
>
> Re: zero-shot performance
> Although not directly discussed in the paper, the zero-shot performance can be seen by looking at the first iterations of the training graphs. In most cases there is some zero-shot performance, as we hope would be the case, given the transfer we are hoping to achieve between tasks. However, further training greatly improves the performance of each of the tasks.
>
> Re: network architecture, network size
> We focused on designing the PLAiD method and minimal architecture tuning was performed. The network architecture is based on that used in the following paper: “Learning Locomotion Skills Using DeepRL: Does the choice of action space matter?”.
>
> Re: Figure 1(c) confusion
> Performance is maintained across all previous tasks (and not just the two most recent). We will clarify this in the text.

---

### Official Review · AnonReviewer4 · 2017-12-08
**Interesting**

**Rating:** 5
**Confidence:** 3

**Review:**

Hi,

This was a nice read. I think overall it is a good idea. But I find the paper lacking a lot of details and to some extend confusing.
Here are a few comments that I have:

Figure 2 is very confusing for me. Please first of all make the figures much larger. ICLR does not have a strict page limit, and the figures you have are hard to impossible to read. So you train in (a) on the steps task until 350k steps? Is (b), (d),(c) in a sequence or is testing moving from plain to different things? The plot does not explicitly account for the distillation phase. Or at least not in an intuitive way. But if the goal is transfer, then actually PLAID is slower than the MultiTasker because it has an additional cost to pay (in frames and times) for the distillation phase right? Or is this counted.

Going then to Figure 3, I almost fill that the MultiTasker might be used to simulate two separate baselines. Indeed, because the retention of tasks is done by distilling all of them jointly, one baseline is to keep finetuning a model through the 5 stages, and then at the end after collecting the 5 policies you can do a single consolidation step that compresses all. So it will be quite important to know if the frequent integration steps of PLAID are helpful (do knowing 1,2 and 3 helps you learn 4 better? Or knowing 3 is enough).

Where exactly is input injection used? Is it experiments from figure 3. What input is injecting? What do you do when you go back to the task that doesn't have the input, feed 0? What happens if 0 has semantics ?

Please say in the main text that details in terms of architecture and so on are given in the appendix. And do try to copy a bit more of them in the main text where reasonable.

What is the role of PLAID? Is it to learn a continual learning solution? So if I have 100 tasks, do I need to do 100-way distillation at the end to consolidate all skills? Will this be feasible? Wouldn't the fact of having data from all the 100 tasks at the end contradict the traditional formulation of continual learning?

Or is it to obtain a multitask solution while maximizing transfer (where you always have access to all tasks, but you chose to sequentilize them to improve transfer)?  And even then maximize transfer with respect to what? Frames required from the environment? If that are you reusing the frames you used during training to distill? Can we afford to keep all of those frames around? If not we have to count the distillation frames as well. Also more baselines are needed. A simple baseline is just finetunning as going from one task to another, and just at the end distill all the policies found through out the way.  Or at least have a good argument of why this is suboptimal compared to PLAID.

I think the idea of the paper is interesting and I'm willing to increase (and indeed decrease) my score. But I want to make sure the authors put a bit more effort into cleaning up the paper, making it more clear and easy to read. Providing at least one more baseline (if not more considering the other things cited by them).

---

> ### Public Comment · (anonymous) · 2017-12-28
> **Thank you for the thorough review, questions, suggestions, and shared interest in the problem at hand!**
>
>   We believe that there is much to be explored for progressive learning and distillation of continuous action tasks, as exemplified by our control problems.
>
> Re: Figure 2
> We will make these figures larger and enhance the explanations.
> As suggested, they do represent a sequence, moving from (a) to (b) to (c) to (d).
> The cost of distillation is accounted for by giving an equal number of simulation time steps to both PLAiD and the MultiTasker. For example if we give the MultiTasker 300k iterations, for PLAID we may use 250k for transfer and 50k for distillation. We show this in Figure 2 by colouring the TL phase in green and the distillation phase in red.
>
> Re: Frequent distillation vs fine-tuning a model through all stages plus a final distillation step
>
> This is an excellent idea for an additional baseline, and we are currently running this “Successive Transfers then Distill” baseline. We will notify the reviewers of the results, as well as update the paper. The results for this new baseline thus far look very similar to that of the MultiTasker; it has a more difficult time learning the additional tasks. We also note that we seek a method that would work when the agent is given new tasks it does not know are coming. For example, after learning to walk on “incline” terrain, the agent does not know about the next three (steps, slopes and gaps), but we want the agent to best prepared to learn it’s next skill, whatever it may be.
>
> Re: details on input injection
> We have added further details and explanations to the paper. Specifically, the “flat” walking is not provided with information about the upcoming terrain, while other policies (e.g., incline, steps, slopes, gaps) are provided with extra inputs (a linear height map) of the upcoming terrain.  Input injection allows for the “flat” walking policy to be used as the starting point for policies that have these additional inputs.
>
> Re:  role of PLAID: continual learning  vs  multitask-solution with maximum transfer
> We view PLAID as a continual learning method, in that we consider the problem of not knowing all tasks beforehand and want to learn any new task as easily/quickly as possible.
> However, it is also proves surprisingly effective as a multitask solution, given the three specific benchmarks that we compare against.

---

### Author Response · Authors · 2018-01-05
**Summary of revisions, and additional baseline**

We have uploaded a revised version of the paper. Changes are currently highlighted in green.
Here is a summary of the changes:
(1) Clarified our goal for PLAID as a continual learning method, while also evaluating its effectiveness as a multi-task solution method, and comparing to multi-task benchmarks.
(2) Updated Figure 2 making it more readable and understandable.
(3) Added text pertaining to noted related work by reviewers.
(4) Clarified how the distillation method is used in PLAiD with DAGGER
  (a) At most 2 experts are used over a set of tasks
  (b) Start by selecting actions from the expert policies and anneal the probability down, leading to more actions being selected
   from the new student policy
(5) Clearer description of why and how feature injection is used.
  (a) Included a diagram showing how and where the new network parameters are added into the network.
  (b) This injection is performed when the policy is learning how to differentiate between the flat and incline tasks.
  (c) Additional figure visualizing the state features.
(6) TL_Only (fine-tuning) comparison -- see Section 8.4
  (a) Ran an additional baseline (average over 5 runs) where TL is done sequentially between tasks WITHOUT distillation.
  (b) Found that TL_Only can also learn new tasks quickly, however the TL_Only method suffers from increased forgetting of
   previously learned tasks. The final distillation to merge all the expert policies together proves to be much more challenging
   than the use of simpler, progressive distillations.
  (c) This method can be considered a version of PLAiD where tasks are learned in groups and after some number of tasks a
     collection of policies/skills are distilled together.
  (d) Still to be done:  adding the TL_Only baseline to Figure 1.
(7) Multiple runs
  We have completed 5 runs for TL_Only and PLAID, with no surprises. The remaining two baselines are still in progress.

---

### Author Response · Authors · 2018-01-20
**More Updates**

We updated figure 1 in the paper to show a diagram for the other added baselines in this work (fine-tuning).
The figures in the paper have been updated, showing the results over 5 runs of each method. The findings are very similar.
Table 1 has also been updated with data for the MultiTasker. This table now better illustrates the effects distillation on "forgetting".

---

### Decision · Program_Chairs · 2018-01-29
**ICLR 2018 Conference Acceptance Decision**

**Decision:**

Accept (Poster)

**Comment:**

The authors propose an architecture that uses a curriculum and multi-task distillation to gain higher performance without forgetting. The paper is largely a smart composition of known methods, and it requires keeping data from all tasks to do the distillation, so it is not truly a scalable continual learning approach. There were a lot of concerns about clarity in the manuscript, but many of these have been assuaged by an update to the paper. This is a borderline paper, but the author's rebuttal and update probably tip it towards acceptance.